# Effect of Caffeine Consumption on the Risk for Neurological and Psychiatric Disorders: Sex Differences in Human

**DOI:** 10.3390/nu12103080

**Published:** 2020-10-09

**Authors:** Hye Jin Jee, Sang Goo Lee, Katrina Joy Bormate, Yi-Sook Jung

**Affiliations:** 1College of Pharmacy, Ajou University, Suwon 16499, Korea; hjjee@ajou.ac.kr (H.J.J.); cw4646@naver.com (S.G.L.); katbormate96@gmail.com (K.J.B.); 2Research Institute of Pharmaceutical Sciences and Technology, Ajou University, Suwon 16499, Korea

**Keywords:** caffeine, neurological and psychiatric disorders, sleep disorder, stroke, dementia, depression, sex differences

## Abstract

Caffeine occurs naturally in various foods, such as coffee, tea, and cocoa, and it has been used safely as a mild stimulant for a long time. However, excessive caffeine consumption (1~1.5 g/day) can cause caffeine poisoning (caffeinism), which includes symptoms such as anxiety, agitation, insomnia, and gastrointestinal disorders. Recently, there has been increasing interest in the effect of caffeine consumption as a protective factor or risk factor for neurological and psychiatric disorders. Currently, the importance of personalized medicine is being emphasized, and research on sex/gender differences needs to be conducted. Our review focuses on the effect of caffeine consumption on several neurological and psychiatric disorders with respect to sex differences to provide a better understanding of caffeine use as a risk or protective factor for those disorders. The findings may help establish new strategies for developing sex-specific caffeine therapies.

## 1. Introduction

Caffeine (1,3,7-trimethylxanthine), a type of methylxanthine series alkaloid [1], is commonly found in coffee, tea, and soft drinks, and also exists in cocoa, chocolate, and a number of dietary supplements [2]. Caffeine is commonly taken orally in the form of coffee or tea, and 99% of it is absorbed into the bloodstream from the gastrointestinal tract, reaching peak concentrations 30–60 min after ingestion and circulation throughout the body [3]. In the USA, adults consume an average of 179 mg of caffeine daily, which is equivalent to 2 cups (100 mg/240 mL) of ground coffee [4]. Caffeine action is thought to be mediated via several mechanisms: the antagonism of adenosine receptors, the inhibition of phosphodiesterase, the release of calcium from intracellular stores, and the antagonism of benzodiazepine receptors [5]. There are also reports that caffeine changes estrogen levels in women [6]. Since estrogen has a neuroprotective or neurotrophic effect and regulates the dopamine system of the black striatum [7], estrogen regulates the effect of caffeine on the dopamine system and suggests that a complex interaction between caffeine, estrogen, and dopamine exists in the basal ganglia system [8]. Furthermore, caffeine is a stimulant for the central nervous system that can penetrate biological membranes, including the blood–brain barrier and placental barrier, and it maintains arousal function in the brain as a nonspecific potent inhibitor of the A1 and A2A adenosine receptors that promote drowsiness [9]. Caffeine also has psychostimulant effects via modulation of the dopaminergic neuron [10], contributing to an attenuated risk for depression in coffee drinkers [7]. Many people consume caffeine to overcome headaches, owing to its vasoconstrictive properties restricting blood flow in the brain [11]. However, excessive caffeine consumption (1~1.5 g/day) can cause caffeine poisoning (caffeinism), which includes symptoms such as anxiety, agitation, insomnia, gastrointestinal disorders, tremors, and mental disorders [12]. Furthermore, depending on the sensitivity, in rare cases, it can also cause death [13]. Caffeine resistance and the rate of caffeine metabolism vary greatly from person to person, especially depending on the activity of the cytochrome P450 1A2 (CYP1A2) gene, encoding an enzyme that breaks down caffeine [6]. CYP1A2 is a major enzyme responsible for the metabolism of purine alkaloid (1,3,7-trimethylxanthine), a caffeine that occurs naturally in coffee beans, and plays an important role in the metabolism of estrogen and coffee [14]. Additionally, depending on endogenous and exogenous factors, the half-life of caffeine is 2 to 10 h (average 3.7 h), and is mainly excreted by urine after being metabolized in the liver [7]. Accumulating evidence has shown sex-specific differences in the activity and expression of many CYP isoforms [15,16,17]. Recent studies demonstrate that CYP1A2 and CYP2E1 activities are higher in men than in women, while the activity of CYP3A, one of the most clinically relevant CYP isoforms, is greater in women [15]. The activity of several other CYP (CYP2C16, CYP2C19, and CYP2D6) isozymes and the conjugation (glucuronidation) activity involved in drug metabolism are higher in men than in women [17]. According to the World Health Organization (WHO), about 6.8 million people worldwide die each year from various neurological and psychiatric disorders, including stroke, Alzheimer’s disease (AD), Parkinson’s disease (PD), and depression. Neurological and psychiatric disorders are not only expensive to treat, but patients have experienced serious stigma, social exclusion, and poor quality of life as a result of their affliction [18]. Over the years, caffeine has been investigated as a potential risk or protective factor for neurological and psychiatric disorders [19]. Some studies have shown that by drinking more than three cups of coffee a day, caffeine reduces the risk of developing AD and PD [20]. Meanwhile, the risk of developing anxiety and panic disorder has been reported to increase after consumption of more than six cups of coffee a day [21]. Interestingly, there are significant sex/gender differences in the prevalence or incidence of neurological and psychiatric disorders. Moreover, a recent study found that individuals with reduced CYP2D6 activity due to the mutated CYP2D6 * 4A (allelic variants of CYP2D6) genotype had a 2.5 times higher risk of PD than those with wild type, which was higher in men [22]. In addition, the presence of high-risk alleles in both CYP17 and CYP19 increased the risk of AD in menopausal women by almost four times [23,24]. From these studies, it has been suggested that the effects of caffeine on the prevalence/incidence of neurological and psychiatric disorders may vary depending on sex, but it is still not thoroughly understood. In the present review, we first review the sex differences in the prevalence and/or incidence of several neurological and psychiatric disorders. Further, this review summarizes the effect of caffeine intake as a risk or protection factor for these disorders in men and women.

## 2. Sex Differences in the Prevalence/Incidence of Neurological and Psychiatric Disorders

Millions of people worldwide are affected by neurological and psychiatric disorders. More than six million people die each year from stroke, and there are 7.7 million new cases of dementia each year [25]. The prevalence/incidence of several neurological and psychiatric disorders in men and women are discussed in the following sections and summarized in Table 1.

### 2.1. Stroke

Stroke is a disease in which the vessels that supply blood to the brain develop abnormalities and suddenly cause local brain dysfunction, accompanied by various neurological deficits such as consciousness disorders, unilateral paralysis, and/or speech disorders [26]. There are two main types of stroke: ischemic stroke, due to lack of blood flow (85%), and hemorrhagic stroke, due to bleeding (15%) [27,28,29,30]. It has been identified that the symptoms of stroke are different between men and women. Fatigue (women vs. men = 31.2% vs. 21.1%), disorientation (44.4% vs. 34.7%), and fever (12.1% vs. 5.3%) appear predominantly in women, while paresthesia (24.2% vs. 37.9%) and ataxia (61.4% vs. 74.7%) are more common in men [31]. In a systematic review by Appelros et. al, it was shown that there was high variance between age groups and countries, but on average, both the incidence and the prevalence of stroke were higher in men than in women [32]. On the other hand, mortality from stroke was greater in women (24.7%) than in men (19.7%). A prospective study from the USA conducted on 505 patients with first ischemic stroke (ischemic stroke genetics study) found that 270 patients (55%) were men and 229 (45%) were women [33]. In their study, no sex differences were found in stroke severity, stroke subtype, or infarct size and location, but a higher percentage of mortality was shown in women [27,28,29,30,33]. In 2009, a study conducted by the Beth Israel Deaconess Medical Center in the USA analyzed 1107 inpatients aged 21 and older who were diagnosed with neuro-ischemic stroke [28]. This study revealed no difference in the prevalence of stroke between men and women, but when comparing the age of patients, women were older than men and were more likely to have heart embolism [28]. Taken together, at most ages, women have a lower or similar risk of stroke than men. However, possibly due to the longer lifespan of women, the incidence of stroke in women gradually increases with age, and as a result, mortality rates in women are higher.

**Table 1 nutrients-12-03080-t001:** Sex differences in the prevalence/incidence of selected neurological and psychiatric disorders.

Diseases	Note	Sex Difference in Incidence/Prevalence	Age	Case Number	Ref.
Stroke	No sex differences in the prevalence of stroke, but women are more likely to have heart attacks and embolism.	M = F	~73	1107	[28]
	No sex differences were found in stroke incidence, severity, or infarct size and location, but female mortality was higher.	M = F	19–94	505	[33]
	Stroke prevalence between ages of 65 and 85 is 41% higher in men than women, and the male/female prevalence ratio decreases with age.	M > F	65–85	30,414	[32]
	Although the incidence of stroke by age is higher in men than in women, the death rate from stroke each year is higher in women because women live longer and have the highest mortality rate at the oldest age (≥85 years).	M > F	56~	1136	[34]
Sleep disorder	Women over 65 have the highest risk of insomnia and have been reported to have increased risk of insomnia as life expectancy is longer in women than in men.	M < F	18~	4885	[35]
	Insomnia symptoms of two nights or more per week are reported in 30.5% in women and 24.5% in men, and for chronic insomnia, the incidence is higher in women (12.9%) than men (6.2%).	M < F	20–35	1395	[36]
	Women are more than twice as likely to be diagnosed with insomnia as men.	M < F	19~	817	[37]
	The diagnosis of insomnia was 9.0% for women and 5.9% for men.	M < F	20~100	1741	[38]
Dementia	A substantially larger number of women than men have AD worldwide.	M < F	65~	NA	[39]
	Rate of progression from MCI to AD was similar in men and women aged 70–79, but higher in women than men after age 80.	M < F	70~	4398	[40]
	In adults over 65, the risk of AD in women is twice as high as in men.	M < F	65~	2611	[41]
	Two-thirds of patients with AD are women.	M < F	65~	5976	[42]
Parkinson’s disease	Incidence rates were consistently higher in men than in women at all ages for PD.	M > F	~90	NA	[43]
	Men had a risk of developing PD twice that of women.	M > F	65~84	4341	[44]
	Women showed higher cognitive abilities than men.	M > F	~80	1741	[45]
	PD is more common in men than women, with an approximate ratio of 2:1.	M > F	19~	902	[46]
Depression	The incidence of more severe depression is higher in women.	M < F	39~65	100	[47]
	In the HCV-infected female population, anxiety and depression were more common than in men.	M < F	41~62	38	[48]
	Greater risk for depression among women compared to men.	M < F	~60	2824	[49]
Anxiety	The incidence of the trait of anxiety is high in women.	M < F	20~23	108	[50]
	Stress-induced anxiety is higher in women than in men.	M < F	19~50	96	[51]
Neuromuscular disease	The incidence of MG was significantly higher in women under age 40, but higher in men over age 50.	M < F	~40	1976	[52]
		M > F	50~		
	Women with CMT1X have less severe consequences for almost all parameters of MNCS compared to men with CMT1X.	M > F	18–79	107	[53]
	The incidence and prevalence of ALS are greater in men than in women.	M > F	~30	NA	[54]

NA, not analyzed; Ref., reference; AD, Alzheimer’s disease; MCI, mild cognitive impairment; PD, Parkinson’s disease; HCV, hepatitis C virus; MG, myasthenia gravis; MNCS, motor nerve conduction studies; CMT1X, Charcot–Marie–Tooth type 1X; ALS, amyotrophic lateral sclerosis; F, female; M, male.

### 2.2. Sleep Disorder

Sleep mediates changes in various physiological functions, including brain activity, breathing, and heart rate, and sufficient sleep improves attention, creativity, memory, and learning [55]. Insufficient sleep or poor sleep quality can act as a risk factor for a variety of diseases, including dementia, psychosis, and diabetes [56,57]. Prevalence of sleep disorders is high, with about 25–30% of population worldwide having some form of inadequate sleep. Sleep disorders degrade the quality of life due to secondary psychological stress as well as promoting physical illness [58]. A recent study found that there was a difference in the prevalence of sleep disorders between sexes. Insomnia, the most common type of sleep disorder, is defined as a condition where it is difficult to initiate and maintain sleeping, resulting in difficulty in early rising [59]. Regarding sex differences in the prevalence of insomnia, many studies have reported that insomnia occurs more frequently in women [37]. In the USA, insomnia diagnosis is double in women compared with that in men, and insomnia symptoms for two nights or more per week have been reported to occur in 30.5% of women and 24.5% of men. In the case of chronic insomnia, the incidence rate was higher in women (12.9%) than in men (6.2%) [36]. In particular, women over 65 have the highest risk for insomnia and have been reported to have an increasing risk with age [35,38]. One of the reasons that women are more susceptible to insomnia than men is the changes in body hormones due to menstruation and menopause [60]. This is because estrogen, an important female hormone, decreases and body symptoms such as hot flashes and sweating are caused by an imbalance of hormones in the body. 

### 2.3. Dementia

Dementia is a pathological neurodegenerative process characterized by a gradual decrease in cognitive, memory, and functional capacity that is severe enough to affect daily functioning [61]. Other symptoms include emotional problems, speech problems, and decreased motivation [62]. AD is the most common form of dementia and most studies do not distinguish AD from all-cause dementia [63]. Global estimates on the prevalence of dementia are up to 7% of the population aged 65 and over, and in developed countries with a longer lifespan, the prevalence is slightly higher still (8–10%) [64]. According to the World Alzheimer’s Report 2015, there are currently 46.8 million people with dementia worldwide, with an estimated increase to 74.7 million by 2030 and 131.5 million by 2050 [65]. Age is a major risk factor for AD, and on average, women live longer than men. However, the difference in lifespan between men and women does not fully explain why two-thirds of Alzheimer’s patients are women. Even after accounting for differences in longevity, some studies have found that women are still at a higher risk [66]. Recently, sex-related differences in neuroanatomy and function are being considered in patient diagnostics, and sex can be an important factor in stratified and personalized treatment in AD patients [39]. Consistent with this finding, analysis of longitudinal data from the Alzheimer’s Disease Neuroimaging Initiative cohort showed that women had greater hippocampal atrophy and faster cognitive decline in the presence of AD biomarkers (Cerebrospinal fluid levels Aβ1-42 and total tau) compared to men [67]. Similarly, a study published in 2017 showed that in dementia patients who were classified as fast progressors, there was a faster rate of dementia in women than men, even when the diagnostic biomarker levels were similar [68]. Sex-related differences and treatment responses related to disease progression after AD diagnosis were also reported [40]. According to a Mayo Clinic study on aging, the progression from mild cognitive impairment (MCI) to AD was similar in men and women in the ages of 70–79, but higher in women than men after 80 years of age. This is likely due to the difference in brain anatomy between men and women, and it is reported that men are expected to withstand more pathologies because their heads are about 10% larger and have more brain volume compared to women, a hypothesis that was supported by autopsy. At the same level of pathology, the probability of clinical diagnosis of AD was found to be significantly higher in women than in men [69]. In the Framingham Study cohort, a study conducted in individuals aged between 65 and 100 years old, incidence of AD in women was twice as high as men [41], and another study reported that two-thirds of AD patients are women [42]. Overall, women showed higher incidence and prevalence of dementia than men, possibly due to various factors, such as longer life expectancy of women and different neuroanatomical function [42].

### 2.4. PD

PD is one of the neurodegenerative disorders with characteristic features, such as hand tremor, muscle stiffness, and postural instability [70]. PD is the second most frequent age-related neurodegenerative disorder, affecting about three percent of people over 65 and five percent over 85 years old [71]. The formation of the Lewy body (α-synuclein accumulation in neurons) in the stromal nigra pars compacta leads to basal ganglion circuit degeneration [46]. Patients under 40 years of age are rare, and prevalence increases with age, approaching three percent of the population over the age of 80 [72]. Increasing evidence has suggested that sex is an important factor in the development of PD. In PD, the onset age, severity, and type of symptoms vary by sex. According to several studies, the onset of PD in men occurs, on average, two years earlier than in women, and the incidence rate in men is twice as high as that in women [73]. It has also been reported that sex differences in PD are determined by the nigrostriatal dopamine system arising from genetic, environmental, and hormonal effects. Sex itself is a variable that can affect the manifestation of non-motor symptoms in PD patients [46]. Women have better cognitive performance than men in two measures: the Symbol Digit Modalities Test, a screening test for cognitive impairment, and Scales for Outcomes of Parkinson’s disease-cognition, a measure of memory and learning, attention, executive function, and virtual space function [45]. Despite the higher incidence of PD in men at all ages, the difference in PD risk between men and women is reduced with age. In those aged 65 to 69, the incidence of PD was shown to be similar between men and women [44]. The reason is likely that women have a longer lifespan than men, and men are at greater risk of dying at a younger age [43]. In addition, motor improvement after deep brain stimulation is similar in men and women, but women are likely to show better improvement in daily living activities compared to men [74]. One of the reasons why the onset of PD is higher in men than in women may be due to the effect of estrogen on dopaminergic neurons and pathways in the brain [75].

### 2.5. Depression

Depression is a common and serious mental disorder that can have long-term consequences and affects all aspects of life. People with depression tend to feel sad, anxious, hopeless, irritable, and ashamed [76]. Severe cases of depression can lead to loss of appetite, sudden weight loss, sleeping problems, and frequent thoughts of death or suicide [77]. It is commonly comorbid with other chronic illnesses and/or mood disorders that make it a complicated disorder difficult to properly diagnose and treat [78]. Depression is more frequently experienced by women compared to men, with a peak in prevalence occurring in middle age. Gender differences in depression are known to be affected by several factors, such as biological, psychological, and environmental factors [79]. In 2018, in Canada, the THINC-integrated tool (THINC-it), a newly developed cognitive tool, was used to evaluate cognitive impairment in patients suffering from major depressive disorder (MDD). It was reported that women had a higher rate of severe depression than men [47]. Additionally, patients with chronic liver disease have a higher incidence of depression than the general population and depression is a common psychiatric comorbidity among individuals with hepatitis C virus (HCV) [80]. Studies have shown that 23% of women, but only 4.1% of men, with chronic HCV have depression. In conclusion, in those with chronic HCV infection, anxiety and depression were more common in women than in men [48]. The University of Michigan’s survey center conducted a community-based study named “American Changing Lives (ACL)” that included two sets of data collected in 1986 and 1989. These data revealed that stressed women were more prone to depression than stressed men [49]. Recent evidence suggests that changes in ovarian hormone levels, especially biological factors such as decreased estrogen, may contribute to increased risk for depression in women [81]. 

### 2.6. Anxiety

Anxiety, which manifests as a sudden increase in alertness, excessive fear, and worry, is the most common mental health disorder and 1 in 9 individuals have experienced anxiety for a year. It is also known that women have a higher prevalence of anxiety than men [82,83]. Results of the State Trait Anxiety Inventory (STAT) score, a psychological inventory that determines individual anxiety and trait anxiety among healthy men and women volunteers at Utretch University Campus, confirmed that women have a high level of trait anxiety [50]. In 2017, a research team at Yale University in the USA conducted an experiment on stress-induced anxiety disorder in healthy adults between the ages of 19 and 50. Their results show that women are more susceptible to stress-induced anxiety [51]. That is, as a result of various anxiety measurement experiments, the incidence of anxiety was found likely to be higher in women than in men. Women have a higher incidence of anxiety disorders not only because they are more sensitive to the lower levels of hormones that make up the stress response, but also because women experience residual anxiety from sexual abuse/violence more often than men [84].

### 2.7. Neuromuscular Disease 

Neuromuscular diseases are a broadly defined group of disorders that involve injury or dysfunction of the peripheral nerve or muscle and include wide variety of disorders, such as multiple sclerosis (MS), Charcot–Marie–Tooth (CMT) disease, amyotrophic lateral sclerosis (ALS), myasthenia gravis (MG), and neuropathic pain [85]. The most common of these diseases is MG, which is an autoimmune disease where the immune system produces antibodies that attach themselves to the neuromuscular junction and prevent transmission of the nerve impulse to the muscle [86]. The onset of MG occurs at any age, but significantly earlier in women than men. The incidence of MG has been reported to be significantly higher in women under age 40, but higher in men over age 50 [52]. CMT disease encompasses a group of disorders called hereditary sensory and motor neuropathies, which damage the peripheral nerves [87]. The highest prevalence of CMT disease occurs at ages 50–64, with men having a higher prevalence than women [53,88]. A study by Nivedita U. Jerath et al. reviewed the results of electrodiagnostic retrospectively in 45 women and 31 men. As a result, women with CMT1X have less severe outcomes for almost all parameters of motor nerve conduction studies (MNCS) (compound motor action potential amplitude, delay time on exercise, and conduction rate) compared to men with CMT1X [53]. ALS is a highly debilitating disease caused by progressive degeneration of motoneurons [89]. Both the incidence and the prevalence of ALS are greater in men than women. The reasons for the difference in the incidence of ALS between men and women is known as the differences in biological responses to exogenous toxins, various exposures to environmental toxins, and fundamental differences between male and female nervous systems and their ability to repair damage [54]. The prevalence/incidence of neuromuscular disease varies according to the age of men and women, but in most cases, it is higher in men than women. The difference in the incidence of ALS between men and women may be explained by differences in the biological response to exogenous toxins. 

## 3. Effect of Caffeine Consumption on the Risk for Neurological and Psychiatric Disorders in Men and Women

According to a paper published in British Medical Journal (BMJ) in 2017, drinking 3–4 cups of coffee per day is associated with a reduced risk of various neurological disorders, including AD, PD, and depression [90]. However, few studies have been reported on sex differences in the effects of caffeine on neurological and psychiatric disorders. This review investigates the differential effects of caffeine on the incidence of symptoms of several neurological and psychiatric disorders in men and women (summarized in Table 2).

### 3.1. Stroke

According to the WHO, 15 million people worldwide suffer from stroke each year, five million of these people die, and another five million are permanently disabled [91]. The causative or protective effect of caffeine on stroke onset has been controversial, and moreover, sex differences have not been studied. In 2015, the National Health and Nutrition Examination Survey in the USA examined the association between coffee consumption and stroke in 19,994 participants (men 9374; women 10,620) over the age of 17. Multivariate analysis found that higher coffee consumption (≥3 cups/day) reduced the incidence of stroke [92]. In 2017, data from the Health Examinees study, a large, prospective, community-based cohort study, were used to analyze the association between coffee consumption and stroke. A survey of about 15,000 men and women between 40 and 69 years of age did not show any significant association between coffee consumption and stroke risk among men. However, in the case of young women, the inverse relationship between coffee consumption and stroke risk was prominent. In other words, higher coffee consumption was found to be inversely proportional to the incidence rate of stroke in women [93]. Some studies show that coffee consumption temporarily increases the risk of ischemic stroke. Mostofsky’s study showed that the incidence of stroke temporarily doubled in those who drank seven or more cups of coffee per week compared with non-drinkers, but there were no gender differences [94]. In summary, the effect of coffee intake on the risk for stroke showed controversial results, but more studies have shown that women have a lower risk of stroke incidence by caffeine intake, than men. As indirect evidence of the preventive effect of coffee consumption on stroke occurrence, there are papers reporting the preventive effect of coffee consumption on the onset of diabetes by maximizing insulin sensitivity, which is a risk factor for stroke, but no differences between sexes were revealed [95].

**Table 2 nutrients-12-03080-t002:** Effect of caffeine consumption on the risk for selected neurological and psychiatric disorders in men and women.

Disease	Note	Risk for Neurological Disorder			Age	Case Number	Coffee Consumption	Ref.
		Men	Women	N.S.				
Stroke	The risk of temporary ischemic stroke increases for an hour after coffee consumption.			+	54~72	390	7 cups/week	[94]
	Higher daily coffee consumption and potential protection from strokes.	-	-		17~	19,994	≥3 cups/day	[92]
	Coffee consumption may modestly reduce risk of stroke.		-		55~	1800	≥4 cups/day	[96]
	Higher coffee consumption among middle-aged Korean women may have protective benefits with regard to stroke risk.		-		40~69	173,357	≥3 cups/day	[93]
Sleep disorder	Middle-aged sleep is more sensitive to increased caffeine dosage than young adults.			+	20~30 40~60	77	≥3 cups/day	[97]
	Caffeine decreased sleep efficiency, sleep time, slow-wave sleep, and REM sleep during the weekly recovery sleep.			+	20~30 45~60	24	165~205 mg/day	[98]
	Adolescent students who consumed high caffeine suffered higher sleep disturbances.			+	12~15	191	52.7 mg/day	[99]
	Short sleep is associated with more caffeine consumption, suggesting that adults with poor sleep quality consume more caffeine.	+	+		19~94	80	164.9 mg/day	[100]
	Habitual coffee intake decreases the efficiency and quality of sleep.	+	+		60~94	162	≥60 cups/year	[101]
Dementia	Moderate regular coffee consumption can have a neuroprotective effect on MCI.			-	65~84	1445	1–2 cup/day	[102]
	An inverse relationship exists between caffeine intake and the risk of dementia.			-	65~	587	200 mg/day	[103]
	Moderate coffee consumption in middle aged individuals may reduce future risk of dementia/AD.	-	-		65~79	1409	3–5 cups/day	[104]
	Elderly women with high caffeine consumption are less likely to have dementia or cognitive impairment.		-		65~	6467	261 mg/day	[105]
	Caffeine appear to reduce cognitive decline in women, especially at higher ages.		-		65~	7017	>3 cups/day	[106]
	Lifetime coffee consumption was positively associated with cognitive performance in elderly women, but not in elderly men.		-		50~	1528	≥3 cups/day	[20]
Parkinson’s disease	The PD risk decreased significantly before 3 cups/day, whereas it did not change materially after 3 cups/day of coffee consumption.			-	65~	5312	3 cups/day	[107]
	Coffee consumption is associated with reduced PD risk in men and women.	- -	-		69~	184,190	2 cups/day	[108]
	Coffee consumption reduces the risk of PD.	-	-		50~79	6710	>10 cups/week	[109]
	A U-shaped relationship exists between caffeine intake and PD in women. Men who consume moderate coffee have a significantly lower risk of PD than men who have never consumed coffee.		+/-		40~75	135,916	1–3 cups/day	[110]
		-						
	The higher the caffeine intake, the lower the incidence of PD in men.	-			45~68	8004	28 oz/day	[111]
Depression	Korean adults who consume caffeine are less likely to become depressed.			-	19~	9576	≥2 cups/day	[112]
	The risk of depression decreases as caffeine consumption increases.		-		30~55	50,739	>4 cups/day	[113]
	Inverse association between caffeine intake and depressive symptoms.		-		18~	5563	309~425 mg/day	[114]
	In secondary school children’s, the effect of caffeine on depression is higher in women than in men.	+	++		11~17	2307	>1000 mg/week	[115]
Anxiety	Anxiety in men increased with increasing doses of caffeine.	+			18~31	99	>150 mg/day	[116]
	In secondary school children, the effect of caffeine on anxiety is higher in males than in females.	++	+		11~17	2307	>1000 mg/week	[115]
Neuromuscular disease	High caffeine intake is significantly associated with a decrease in developing MS.	-	-		18–69	1620	6 cups/day	[117]
	Caffeine intake does not affect the risk of MS in white women.			-	25–42	258	0–5 cups/day	[118]
	People who drink more than one cup of coffee per day for at least 6 months have a lower risk of ALS compared to people who do not drink coffee at all.	-	-		26–94	1031	>1 cup/day	[119]

+: increase, + +: increase to a great extent, -: decrease, - - decrease to a great extent. N.S.: not significant; Ref., reference; REM, rapid eye movement; MCI, mild cognitive impairment; AD, Alzheimer’s disease; PD, Parkinson’s disease; MS, multiple sclerosis; ALS, amyotrophic lateral sclerosis.

### 3.2. Sleep Disorder

Caffeine overdose can delay sleep onset, reduce total sleep time, change normal sleep stages, and reduce sleep quality. Caffeine-induced sleep disorder is known as a psychiatric disorder caused by excessive caffeine consumption [120]. A double-blind cross-design study of 22 young participants (10 men, 12 women; 20–30 years old) and 25 middle-aged participants (12 men, 13 women; 40–60 years old) showed that in terms of sleep volume and efficiency, middle-aged participants in good health were more susceptible to increased caffeine doses compared to young adults [97]. In a study of 12 young and 12 middle-aged subjects who consumed one to three cups of coffee per day, caffeine intake was found to reduce sleep efficiency, sleep time, slow-wave sleep (SWS), and rapid eye movement (REM) sleep in both age groups. However, during the weekly recovery, middle-aged participants had significantly reduced sleep time and sleep efficiency compared to younger participants [98]. In addition, for 66 boys and 125 girls who consumed similar amounts of caffeine, the average sleep time decreased from 528.8 min (8.8 h) on Saturday nights, to 448.5 min on Sunday nights (7.5 h). Perhaps not surprisingly, teenagers who consumed large quantities of caffeine experienced interfered sleep [99]. Furthermore, a survey on the quality of sleep in 26 adult men and 54 women with an average daily caffeine intake of 164.9 mg, showed that 80% of respondents suffered from sleep disturbances once a week [100]. Finally, among 162 cognitively healthy Koreans aged 60–94 (85 men, 77 women), people who consumed more than 60 cups of coffee per year had a 20% lower volume of pineal parenchyma, a melatonin-producing region, than those who consumed less than 60 cups of coffee per year [101]. In summary, caffeine intake negatively affected the quality of sleep and the amount of sleep with age, with no differences seen between men and women.

### 3.3. Dementia 

A number of studies report that caffeine consumption tends to decrease the incidence of dementia. Increased caffeine intake in white women aged 65–80 has been reported to lower the likelihood of dementia or cognitive impairment [105]. In addition, drinking a moderate amount of coffee (3~5 cups/day) lowers the incidence of dementia compared with not drinking coffee, and among coffee-drinkers, the incidence of dementia is lower in women than in men. Moreover, a later-life survey found that low consumers of coffee were more likely to develop depression (based on the Beck depression scale) compared to moderate coffee consumers [104]. According to a study by Vincenzo et al., individuals who habitually consumed moderate amounts of coffee (one to two cups of coffee a day) had a lower incidence of MCI than those who did not drink coffee [102]. A study of 587 people in a California retiree community found that those who consumed more than 200 mg of caffeine per day at the age of 90 and took extra vitamin C significantly reduced their risk for dementia [103]. Some studies have reported neuroprotective effects of caffeine by showing that women with high caffeine intake (more than three cups per day) have fewer speech retrieval and decreased spatiotemporal memory problems than women who consumed less than one cup of coffee per day [106]. The psycho-stimulating component of caffeine appears to reduce cognitive decline in women without dementia, especially in the elderly [106]. Retrospective observational studies have shown that lifetime coffee consumption tends to increase cognitive performance in aged women, but this is not the case in aged men [20]. Taken together, these studies show that caffeine intake did not help improve cognitive abilities in either men or women, although steady caffeine intake seems to reduce the risk of developing dementia for both men and women, with a greater effect in women.

### 3.4. PD

The possible relationship between caffeine intake and PD risk has attracted considerable attention since the early 1970s, and more and more observational studies have been conducted on this [107]. An inverse association between coffee consumption and PD risk has been found in several epidemiological studies [110,121,122,123]. Even though the evidence is increasing that caffeine intake can reduce the risk of PD, the number of cohort studies is still relatively small and is almost exclusively limited to the USA [109]. A meta-analysis of the Hui Qi research team found that consuming less than three cups of coffee per day significantly reduced the risk of PD, while consuming more than three cups of coffee per day did not significantly alter the risk for PD [107]. The link between caffeine consumption and the risk of developing PD was more pronounced in men than women. A study by Ascherio et al. reported that men who consumed a moderate amount of coffee had a significantly lower risk of PD than men who did not drink coffee at all [110]. For women, there is a U-shaped relationship between coffee consumption and Parkinson’s disease risk, with women drinking 1–3 cups of coffee per day having the lowest risk. These results support the protective effect of moderate amounts of caffeine on Parkinson’s disease risk. A cohort study conducted in the USA in 1992 found that in men, regular coffee consumption was associated with a reduced risk of PD [108]. In the case of women, the risk of PD was significantly reduced in the group with the highest caffeine intake (four or more cups per day) compared to the group with the lowest caffeine intake (less than one drink per day), but the decrease was lower than in men. According to a study published in 2000 by Ross et al., the incidence of PD observed among Japanese men participants aged 45 to 68 was two to three times higher in non-coffee drinkers than in coffee drinkers [111]. In summary, high caffeine intake is associated with a protective effect that suppresses PD incidence in men and women, significantly reduces the risk of PD in men, and only slightly reduces the risk in women. These results suggest that men and women respond differently to caffeine administration and that these gender differences may be mediated by changes in circulating steroid hormones [124].

### 3.5. Depression

Adequate caffeine intake has a positive effect on depression, but excessive caffeine intake can exacerbate depression by stimulating sympathetic nerves [125]. Moderate amounts of caffeine also help prevent an imbalance in brain neurotransmitters, such as serotonin and dopamine, that cause depression. The effect of caffeine intake on depression was investigated in students aged 11 to 17 years old (a total of 2307 students). As a result, consuming less than 1000 mg of caffeine per week increased the incidence of depression in girls compared to boys [115]. Unlike the above results, according to a survey conducted by the Centers for Disease Control and Prevention, the prevalence of depression decreased as caffeine intake increased. The incidence of depression among participants who drank more than two cups of coffee per day was reduced by 24% compared to those who didn’t drink coffee [112]. In addition, as a result of analysis of data from the National Health and Nutrition Examination Survey conducted in 2019, it was found that the incidence of depression decreased as the amount of caffeine intake increased, but gender differences were not analyzed [114]. A 2011 cohort result from the US Nurses’ Health Study found an inverse age-adjusted dose-response relationship between caffeine-containing coffee and depression risk in women. Compared with the group with the lowest caffeine consumption (<100 mg/d), the relative risk for depression was lower in the group with the highest caffeine consumption (≥550 mg/d). In other words, women who consumed more caffeine had a lower risk of depression than women who consumed less caffeine [113]. In summary, the risk of developing depression was decreased by caffeine intake to a greater extent in women than in men, and the effect of caffeine intake on depression incidence was different according to the age of women. In particular, in adolescence, caffeine decomposition ability is lower than that of adults, so the staying time in the body is relatively long and it can increase the risk of depression by inducing sleep disorders [126].

### 3.6. Anxiety 

Generalized anxiety disorder is a serious mental illness that affects up to 6% of population in the world. Symptoms are complicated by the consequences of accompaniment with other mental disorders, such as MDD, panic disorder, and alcohol/substance abuse, resulting in worsening of symptoms and poor treatment responses [127]. Excessive caffeine can cause symptoms ranging from general anxiety to compulsive disorders [120]. However, few studies have been conducted on the incidence of anxiety in men and women, by caffeine. As a result from a cohort study of 3323 students aged 11–17 years (boys 48.5%, girl 51.5%), the effect of caffeine on anxiety was not significant in girls, but in boys, anxiety increased with caffeine intake [115]. Consistent with the above results, at the University of Valencia, Spain, a STAT test of 39 men and 60 women between 18 and 31 years of age showed that men had higher state anxiety than women [116]. As shown above, the effects of caffeine on anxiety were more pronounced in men than in women. However, very little research has been conducted to assess the association between caffeine intake and anxiety.

### 3.7. Neuromuscular Disease

There are only a few studies about the effect of caffeine on neuromuscular diseases, and little is known about its sex differences and mechanisms. A case-controlled study from the European ALS Consortium (EURALS Group) reported that people who drink more than one cup of coffee per day for at least six months have a lower risk of ALS than those who didn’t drink coffee at all [119]. Similar findings of caffeine’s impact on the risk of developing MS were found in two cohort studies conducted in the USA and Sweden in 2016 [117]. Compared to those who never had coffee, those with high coffee intakes in excess of six cups per day had a significantly reduced risk of MS in both men and women. However, a recent meta-analysis of five large cohort studies conducted in the USA showed no association between coffee consumption and ALS risk, in both males and female. Another large prospective study conducted by Massa et al. also reported no association between caffeine consumption and MS risk in white women [118]. 

## 4. Conclusions 

This review has shown that the beneficial and/or risky effects of caffeine on several neurological and psychiatric disorders may vary depending on sex. In the case of stroke, caffeine intake has a greater protective effect in women than in men, and for sleep disorders, caffeine intake tends to increase the risk to a similar extent in both men and women. This review also shows that the risk for developing dementia is reduced to a greater extent in women than in men. In contrast, the protective effect of caffeine against PD was found to be greater in men than in women. Notably, in the case of anxiety and depression, the effect of caffeine on the risk for their incidence tends to be age dependent. In fact, the risk of depression has shown to decrease in adult women but not in men, while in adolescence, women have a much higher risk of depression than men. For anxiety, the risk seems to be increased primarily in adult men but not in women, while during adolescence, the risk increases in both men and women, but to a much greater extent in men. In other words, caffeine consumption not only has a positive effect of reducing the risk of stroke, dementia, and depression in women and reducing the risk of PD in men, but also has a negative effect of increasing sleep disorders and anxiety disorders in adolescence in both men and women. Moreover, there are not many research articles that analyzed individual sex/gender differences in the effect of caffeine on neurological disorders. Therefore, further studies focusing on sex/gender differences are needed to fully understand the positive and negative effects of coffee intake on neurological and psychiatric disorders in men and women, and to develop new strategies for sex-specific caffeine use.

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
