# Peer review of "Effect of Caffeine Consumption on the Risk for Neurological and Psychiatric Disorders: Sex Differences in Human"

_nutrients, 2020, doi:10.3390/nu12103080_

Round 1

Reviewer 1 Report

Very concise review

I suggest to focus on the pure neurological disease not only stroke, dementia, parkinson but also neuromusclular disease etc.

The review better to give more information about the mechanism of caffeine in these disease course.  It will be much better if you can explain both the positive and negative effect of caffeine in these results.

Author Response

Point-by-point response to the reviewer’s comments

Reviewer-1

  1. I suggest to focus on the pure neurological disease not only stroke, dementia, Parkinson but also neuromuscular disease etc.

→ Thank you very much for pointing out an important point. In response to your comments, we further investigated gender differences in the prevalence and/or incidence of neuromuscular disorders such as myasthenia gravis, Charcot-Marie-Tooth disease and amyotrophic lateral sclerosis. In addition, we further investigated gender differences in the effects of caffeine intake as a risk or protective factor for multiple sclerosis and amyotrophic lateral sclerosis. The new sentences were added in 2.7 Neuromuscular disease (the lines 213-234 of the revised manuscript) and 3.7 Neuromuscular disease (the lines 369-380 of the revised manuscript), and we also changed Tables 1 and 2 by inserting new category shown below into the Tables.

  1. The review better to give more information about the mechanism of caffeine in these disease course.

→ We agree with your opinion. As your advice, I have added a new sentence on the caffeine mechanism to the lines 29-36 and 47-49 of the revised manuscript.

  1. It will be much better if you can explain both the positive and negative effect of caffeine in these results.

→ We appreciate your comments. As you pointed out, we added new sentence “In other words, caffeine consumption not only has a positive effect of reducing the risk of stroke, dementia, and depression in women and reducing the risk of PD in men, but also has a negative effect of increasing sleep disorders and anxiety disorders in adolescence in both men and women. Moreover, there are not many research articles that analyzed individual sex/gender differences in the effect of caffeine on neurologic diseases issues significantly. Therefore, further study focusing on sex/gender differences is needed to fully understand the positive and negative effect of coffee intake on neurological diseases in men and women, and to develop new strategies for sex-specific caffeine use.” in the lines 392-399 of the conclusion section of the revised manuscript.

Reviewer 2 Report

The authors presented a review article on the Effect of Caffeine Consumption on the Risk for  Neurological Diseases: Sex Differences in Human.

The topic is interesting and well within the aims and scopes of the Journal.

The manuscript was well prepared but I think some further details and explanations must be presented and discussed in some parts of it.

These things are listed below one by one:

ABSTRACT:

- “However, excessive consumption can cause caffeine (caffeinism),…” What’s the meaning of this exactly?

INTRODUCTION:

- “Many people consume caffeine to overcome headaches, owing to its vasoconstrictive properties restricting blood flow in the brain.” Please cite a reference for this.

- “Moreover, recent studies have shown that a decrease in the level of the CYP2D6 gene increases the risk of developing PD [15] and that mutations in the CYP17 and CYP19 genes lead to a four-fold increase in AD risk [16,17].” Please develop this concept a little. For example, decrease to what extent? Mutations at what level?

SECTION 2:

- “Fatigue, disorientation, and fever appear predominantly in women, while paresthesia and angina are more common in men”. Please give the percentages here.

- Line 149: “…the age of 80. [61].” Check this writing.

- All this section: Pease try to give a biological, chemical or medical explanation of all these results where not reported i.e. why one sex is more affected with that problem than the other. This part would really complete your review.

SECTION 3:

- All this section: Pease try to give a biological, chemial or medical explanation of all these results where not reported i.e. why one sex is more affected with that problem than the other. This part would really complete your review.

-“In some studies, the effect of caffeine on anxiety was more pronounced in men than in women, depending on the dose of caffeine.” And these studies are?

Author Response

Point-by-point response to the reviewer’s comments

Reviewer-2

The authors presented a review article on the Effect of Caffeine Consumption on the Risk for Neurological Diseases: Sex Differences in Human. The topic is interesting and well within the aims and scopes of the Journal. The manuscript was well prepared but I think some further details and explanations must be presented and discussed in some parts of it. These things are listed below one by one:

→ Thank you for your kind comments. Agreeing with your advices and suggestions, we have proofread carefully and corrected several points throughout the manuscript.

  1. Abstract: “However, excessive consumption can cause caffeine (caffeinism),…” What’s the meaning of this exactly?

→ Thank you for your kind comments. Excessive caffeine intake generally means 1–1.5 grams per day in large amounts enough to cause caffeine poisoning (Willson 2018). Therefore, we have changed the previous sentence “However, excessive consumption can cause caffeine (caffeinism),” with “However, excessive caffeine consumption (1-1.5 g/day) can cause caffeine poisoning (caffeinism),” in the lines 11-12 and 42-45 of the revised manuscript.

  1. Introduction: “Many people consume caffeine to overcome headaches, owing to its vasoconstrictive properties restricting blood flow in the brain.” Please cite a reference for this.

→ As your comments, we added reference [11] (Addicott, Yang et al. 2009) for this sentence in the line 42 of the revised manuscript.

  1. Introduction: “Moreover, recent studies have shown that a decrease in the level of the CYP2D6 gene increases the risk of developing PD [15] and that mutations in the CYP17 and CYP19 genes lead to a four-fold increase in AD risk [16,17].” Please develop this concept a little. For example, decrease to what extent? Mutations at what level?

→ We agree with your comments and have added more details on the CYP gene that affects AD and PD “Moreover, a recent study found that individuals with reduced CYP2D6 activity due to the mutated CYP2D6 * 4A genotype (allelic variants of CYP2D6) had a 2.5 times higher risk of PD than those with wild type, which was higher in males [21]. In addition, the presence of high-risk alleles in both CYP17 and CYP19 increased the risk of AD in menopausal women by almost 4 times [22,23].” in the lines 66-70 of the revised manuscript.

  1. SECTION 2: “Fatigue, disorientation, and fever appear predominantly in women, while paresthesia and angina are more common in men”. Please give the percentages here.

→ Thanks for your comment. As mentioned, We added the male and female percent of stroke symptoms ”Fatigue (female vs male = 31.2% vs 21.1%), disorientation (44,4% vs 34.7%), and fever (12.1% vs 5.3%) appear predominantly in women, while paresthesia (24.2%vs 37.9%) and ataxia (61.4% vs 74.7%) are more common in men”, which are on lines 86-88 of the revised manuscript.

  1. SECTION 2 Line 149: “…the age of 80. [61].” Check this writing.

→ As your comments, we removed the word “.” in line 162 of the revised manuscript.

  1. SECTION 2: All this section: Please try to give a biological, chemical or medical explanation of all these results were not reported i.e. why one sex is more affected with that problem than the other. This part would really complete your review.

→ We agree with your comments. As your advice, we've revised Section 2 drastically to add why one gender affects the problem more than others. Added new sentences in lines 119-122, 176-178, 197-199, and 210-212 of the revised manuscript.

  1. SECTION 3: All this section: Please try to give a biological, chemical or medical explanation of all these results were not reported i.e. why one sex is more affected with that problem than the other. This part would really complete your review.

→ We appreciate your comments. As your advice we've revised Section 3 drastically to add why one gender affects the problem more than others. Added new sentences in lines 259-262, 330-332 and 353-355 of the revised manuscript.

  1. SECTION 3: “In some studies, the effect of caffeine on anxiety was more pronounced in men than in women, depending on the dose of caffeine.” And these studies are?

→ We appreciate your comments. We recognized errors in the manuscript and revised it with a new sentence to deliver the correct content "As shown above, caffeine's effect on anxiety was more pronounced in men than in women" in lines 366-367 of the revised manuscript.
